# Quantification of porosity in extensively nanoporous thin films in contact with gases and liquids

Netanel Shpigel [1], Sergey Sigalov[1], Fyodor Malchik[1], Mikhael D. Levi[1], Olga Girshevitz[1], Rafail L. Khalfin[2] & Doron Aurbach[1]*

Nanoporous layers are widely spread in nature and among artificial devices. However, complex characterization of extensively nanoporous thin films showing porosity-dependent softening lacks consistency and reliability when using different analytical techniques. We introduce herein, a facile and precise method of such complex characterization by multi-harmonic QCM-D (Quartz Crystal Microbalance with Dissipation Monitoring) measurements performed both in the air and liquids (Au-Zn alloy was used as a typical example). The porosity values determined by QCM-D in air and different liquids are entirely consistent with that obtained from parallel RBS (Rutherford Backscattering Spectroscopy) and GISAXS (Grazing-Incidence Small-Angle Scattering) characterizations. This ensures precise quantification of the nanolayer porosity simultaneously with tracking their viscoelastic properties in liquids, significantly increasing sensitivity of the viscoelastic detection (viscoelastic contrast principle). Our approach is in high demand for quantifying potential-induced changes in nanoporous layers of complex architectures fabricated for various electrocatalytic energy storage and analytical devices.

[1] Department of Chemistry, Bar Ilan Institute for Nanotechnology and Advanced Materials (BINA), Bar-Ilan University, 52900 Ramat-Gan, Israel. [2] Departments of Mechanical Engineering and Chemical Engineering, Technion - Israel Institute of Technology, 32000 Haifa, Israel. *email: Doron.Aurbach@biu.ac.il

Thin porous layers are widely used in many scientific and practical applications[1]. The high surface area associated with porosity of many composite materials provides a great benefit in improving the activity of the catalytic substances, increasing their charge storage capacity and rate capability in supercapacitor-based devices[2], and enhancing sensing capabilities of sensors[3]. The electrical properties of thin porous layers possess an advantage in fabrication of ultralow-$k$ dielectric films for minimization of RC delay in nano-integrated circuits, while the low thermal conductivity of such films can be utilized for engineering of high-performance thermally insulating barriers[4]. Thin mesoporous coatings characterized by low refractive index are highly desirable for optical applications, such as antireflection coatings[5] and optical microresonators[6]. The physical and morphological properties of nanoporous and mesoporous layers are strongly linked to materials' porosity (defined as the ratio of the void volume to the total volume of the layer). Hence, many efforts have been devoted to tuning porosity of nanoporous and mesoporous structures to their applications in various devices. While physical and chemical methods of porous layer syntheses are well developed, a reliable self-consistent quantitative characterization of their porosity, especially, in contact with liquids, has never been reported presenting a significant challenge to nanotechnology field.

The most common techniques to quantify microporosity and mesoporosity of various materials include BET (Brunauer–Emmett–Teller) based on analyses of mass adsorption/desorption isotherms and mercury porosimetry for characterization of porous materials. These techniques, however, are more appropriate for characterization of bulk materials (analyzed masses of more than hundreds of μg/cm$^2$) rather than that of thin films. Over the years, several methodologies were suggested for an accurate determination of porosity in thin layers. The use of radiation-based techniques, such as small angle X-ray scattering (SAXS)[7], grazing-incidence small-angle scattering (GISAXS)[8] and ellipsometry[9] both operating in gas phase (usually in air), or using X-ray/neutron porosimetry[10] in contact with solvent vapors, have been successfully applied. Quantification of porosity can be also done via elemental analyses techniques such as energy-dispersive X-ray spectrometry (EDX)[11] or Rutherford backscattering spectroscopy (RBS)[12] by considering the mass per unit volume of the measured materials, i.e., gravimetric density of the material. In this case, the thickness of the coating should be measured independently.

As a background for the work we performed, which is reported herein, we discuss below several previous works.

As was previously reported,[13] quartz crystal microbalance (QCM) measurements in their simple gravimetric mode were applied to determine porosity of Au films fabricated from Ag (70 at. %)-Au alloy sputtered onto a quartz crystal surface, which was leached in concentrated nitric acid[14]. The porosity was determined by comparison of the experimental specific density of thus obtained porous Au films with the known tabulated value of specific density of non-porous Au. Unfortunately, completeness of the leaching process was not verified in that paper, and hence thus calculated porosity of the Au host is a bit questionable with respect to its reliability. Since the QCM measurements were performed in air, it is not clear whether the highly porous Au host keeps the same porosity and rigidity when immersed from air into liquids since this check would imply the use of QCM-D (QCM with dissipation monitoring) on multiple harmonics instead of conventional QCM (without dissipation monitoring) implemented in this paper.

In other previous studies, ultra-thin organic films[13], proteins[15], and layers of biochemically active species of different nature[16] were measured by a combination of QCM measurements in liquids coupled with independent assessment of dry adsorbent mass or thickness by various optical techniques (such as ellipsometry, atomic force microscopy, surface plasmon resonance, etc). This allowed a calculation of the effective adsorbent layer thickness or its density. QCM-D measurements in air cannot in this case be done because the biochemically active layers of adsorbent species are formed (and exist) in liquid environments. Since the above noted optical techniques reflect dry mass of the adsorbent films, their combination with QCM measurements characteristic of the mass of the wet films allows for a calculation of the amount of solvent chemically or electrostatically bound to the adsorbent or trapped in narrow pores/cavities of the adsorbent layer. Usually the mass or volume fraction of the adsorbent in these layers is a matter of interest rather than the film's porosity[17]. Finally, the attempt of a simultaneous determination of porosity and viscoelasticity changes caused by adsorption of TiO$_2$ nanoparticles onto extensively rough Si host from the related QCM-D measurements in liquid[18] has completely failed for the following two reasons: (i) The combination of viscoelastic and hydrodynamic modeling is valid only for slight (shallow) roughness of the host matrix rather than for host matrices with strong roughness[19]: actually, the host considered in this paper possesses strong roughness; (ii) The authors report that the fitted values of permeability lengths are significantly larger than that of the porous layer thickness itself. This result is meaningless in the framework of the hydrodynamic model[20] that the authors claim to be used in their work. The above discussed previous studies provide excellent driving force for the studies described in this paper.

Herein we propose a facile method for quantification of nanoporosity of thin solid layers based on their multi-harmonic QCM-D characterizations in air (i.e., in their dry state) and in contact with different liquids. As a model system of extensively nanoporous solid films important in the field of catalysis, energy storage, and biosensing devices, we have chosen thin nanoporous Au-Zn alloy film fabricated on the top of a MHz frequency range quartz crystal (QC) sensor by electrochemical alloying–dealloying of Au (with Zn) in Zn$^{2+}$-containing electrolyte solution. The definition of the term extensively nanoporous solid film (literarily denoted sometimes as highly porous film) deserves some explanation, because it depends on the materials' properties and the characterization techniques used. In the context of this paper, an extensively nanoporous Au-Zn alloy film is understood as a material possessing sufficiently high porosity (≈37% in the considered case), which significantly affects the stiffness of the material: high porosity makes solid materials softer. Such a definition automatically excludes the use of conventional QCM instruments (without dissipation monitoring) for solving one of the challenging problem of modern nanotechnology, namely, a real-time monitoring of the changes in materials' porosity together with the related changes in the materials' mechanical properties. This was implemented herein using the principle of viscoelastic contrast when one and the same thin films attached to QC surfaces were characterized by QCM-D both in air and in liquids. Consequently, the porosity was calculated based on both gravimetric and beyond-the-gravimetric QCM-D measurements on multiple harmonics and also using the independent complementary techniques (RBS and GISAXS), demonstrating good agreement between them, see Fig. 1. This paper shows that QCM-D in combination with complimentary techniques can be successfully used as a precise, powerful, and facile method for complex characterization of extensively nanoporous films.

## Results

**Fabrication of thin nanoporous Au-Zn alloy films**. Reliable QCM-D analyses of layers' porosity require deposition of thin nanoporous films on top of the QC surfaces. The coated layers

should form a sufficiently non-slipping contact to the sensor surface in order to simplify the interpretation of the QCM-D characteristics of the porous layer. To meet this requirement, we have fabricated a model system in which the nanoporous layer is formed across the entire depth of the Au-Zn film. A thin Au layer (typically 70-nm thick) serves as a convenient current collector in QCM measurements. In order to ensure good adhesion of the Au layer to the QC surface, the latter is coated by intermediate ultra-thin Cr or Ti coating. The formation of the porous electrodes was carried out by applying potential scanning in the range between 1.5 V and −0.4 V (vs. Zn metal as a reference electrode, using another Zn counter electrode) in a solution of 1.5 M ZnCl$_2$ in ethylene glycol at 120 °C, following the procedure reported elsewhere[21,22]. The potential scanning started from open-circuit potential of 0.6 V, being directed toward the negative potentials. Below 0.1 V, a sharp increase of the cathodic current was observed indicating the initial nucleation of Zn on the Au electrode's surface. At the positive scan at the lowest potentials (<0 V), a typical nucleation loop was observed, associated with an increase of the electrode surface area. Two clearly resolved anodic peaks were recorded assigned to oxidation of Zn from two different forms: the lowest anodic peak is attributed to dissolution of the deposited Zn layer, whereas the higher peak stems from extraction of Zn from the Au-Zn alloy[23]. It is important to note that, according to the theoretical description of this process, only dissolution of Zn occurs while the amount of Au remains constant[24]. However, the entire alloying/dealloying process forms a highly porous Au-Zn thin film. The formation of a nanoporous layer resulted in a change of the Au color as is seen from inset image in Fig. 2.

The morphology of the electrode was examined before and after cycling using scanning electron microscopy (SEM) and atomic force microscopy (AFM). Figure 3a shows the surface of a polished Au electrode characterized by a smooth morphology (typical roughness about 2 nm). The grain boundaries between the Au particles of the pristine electrode (average size 50 nm) can be easily seen. As follows from Fig. 3b, owing to the repeated electrochemical alloying/dealloying process, a porous layer was formed with average pore size of 50 nm. The tilted image of the electrode cross-section (Fig. 3c) displays a step from the QC plate to the porous film demonstrating the inner structure of the pores. While from the top of the projected image the pores are seen as circular holes, from the cross-section, one can observe widening of the voids toward the QC plate forming "bell-shaped" holes. The atomic description of the Zn dissolution and the morphologically associated process described by Erlebacher[25] is in good agreement with our observations. The thickness of the layer was measured by AFM and found to be 180 nm in average as presented in the image (Fig. 3d).

**Determination of nanoporosity of Au-Zn alloy by QCM-D.** Quantification of porosity of nanoporous films by the QCM-D method is based on determination of either area-averaged mass density changes due to formation of the nanoporous layer *measured in air* ($\Delta m_p$, units µg/cm$^2$) or tracking mass density of a liquid rigidly trapped in voids of nanoporous films *measured in liquids* ($\Delta m_{liq}$, units µg/cm$^2$). We first present clear evidence that high quality factor of the QC acoustic sensors is not changed after repeated cycles of electrochemical alloying/dealloying of Au coating of the crystal. In fact, the shape of the resonance peaks for the neat commercial Au-coated QC and for the same QC after the formation of the nanoporous Au-Zn alloy remains unchanged (see Supplementary Fig. 1 and the related description in Supplementary Note 1). Then QCM-D responses of the porous layer characterized by coupled normalized frequency and dissipation

factor changes ($\Delta f/n$, and $\Delta D$, respectively, where $n$ is the overtone order) were measured both in air and in contact with liquid. The essentially gravimetric QCM-D response of a sample has the following property: the changes of $\Delta f/n$ must be $n$-independent, whereas the related values of the dissipation factors, $D$, remain unchanged (i.e., $\Delta D = 0$) for all $n$[26]. This property is characteristic of thin rigid nanoporous layers measured in air (reference state is the neat (uncoated) QC in air) or of liquid rigidly trapped in nanosized pores of the film (reference state is the coated QC in air) quantitatively described by the well-known Sauerbrey equation[27]:

$$\Delta f/n = -C\Delta m \tag{1}$$

where $C$ is sensitivity constant of the QC, which for the fundamental frequency $f_0 = 5$ MHz is equal to 56.6 Hz/(µg cm$^2$)[26]. Note that the change of the dissipation factor, $\Delta D$, related to any overtone order, $n$, is the ratio of the normalized full resonance peak width change, $\Delta W/n$, to the normalized absolute resonance frequency, $f/n$: $\Delta D = (\Delta W/n)/(f/n)$. In particular, for the fundamental oscillation ($n = 1$), the dissipation factor change is reduced to $\Delta D_0 = \Delta W_0/f_0$.

During QCM-D measurements of the attached rigid solid layers in air, the acoustic wave generated by the oscillating QC almost entirely reflects from the layer/air interface forming a standing wave in the coated crystal (see Supplementary Fig. 2a, b); note that the wave velocity across the solid layer is exactly the same as that of the QC surface. In contrast, during QCM-D measurements of the same layer in a liquid, in addition to the formation of the standing wave in the coated crystal, the acoustic wave partially penetrates into the contacting liquid experiencing viscous dissipation on the characteristic lengths called the penetration depth, $\delta$ (see Supplementary Fig. 2c, d), defined by the equation[26,28]:

$$\delta = \sqrt{\frac{\eta_{liq}}{\pi n f_0 \rho_{liq}}} \tag{2}$$

Here $\eta_{liq}$ and $\rho_{liq}$ are dynamic viscosity and specific density of the contacting liquid. It is seen that the shear wave at higher overtone orders, $n$, have shorter penetration depths than that for the lower overtone orders, i.e., the amplitude of oscillation decays on a short distance, close to the external surface of the nanolayer as shown in Supplementary Fig. 2c. In contrast to the measurements in air, viscous dissipation of the oscillation energy makes the changes of $\Delta f/n$ and $\Delta W/n$ to become $n$-dependent and hence affected by viscosity and density of the liquid. However, when the porous layer contains nanopores much narrower than the penetration depth, $\delta$, the liquid inside the nanopores is trapped, hence presenting the inertial load showing contribution to the frequency change only (no additional dissipation change takes place). The mass of the trapped liquid can thus be calculated using the Sauerbrey formula (Eq. (1)). If in contrast much wider pores are present, hydrodynamic correction accounting for the dissipation of oscillation energy in the wide pores is required[29] (a more detailed explanation can be found in Supplementary Note 2).

Measurements in air were performed after seven alloying/dealloying cycles of the Au film. A negative frequency shift of 1.61 kHz was recorded corresponding to additional mass loading of 28.5 µg/cm$^2$ (calculated using Eq. (1)) due to the formation of stable nanoporous Au-Zn phase, which does not dissolve during the oxidation stage. The presence of Zn in the bulk of Au film after its alloying/dealloying was later confirmed by RBS analyses, which gave the similar Zn mass value of 28.8 µg/cm$^2$ as discussed below. Based on the mass density of the added Zn, and considering the initial Au layer 73.5-nm thick (as confirmed by

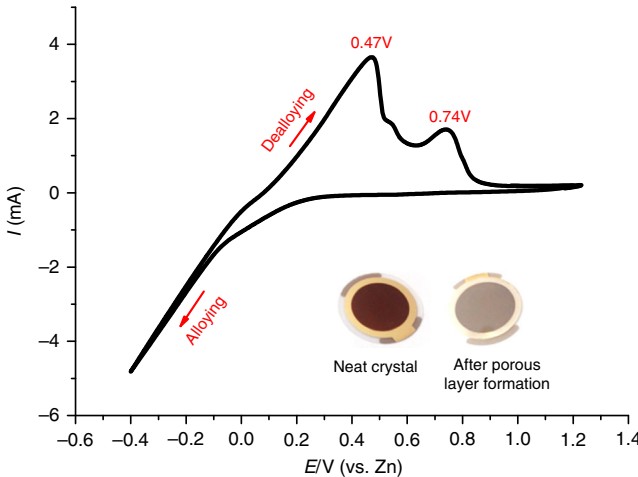

**Fig. 1** Schematic illustration of the proposed approach to assess porosity in nanolayers. **a** Typical structure of nanoporous Au-Zn alloy fabricated on the top of quartz crystal sensor (1 μm scale bar). The neat Au electrode was covered by 1 mm² Kapton tape to form a non-alloyed surface for step measurement by AFM. **b** Graphical summary of all the presented porosity quantification techniques: the profile of the oscillation waves in liquid produced by the vibrating quartz sensor is represented by the black solid line. The GISAXS (grazing-incidence small-angle scattering) technique is represented by incident and scattered X-ray beams (red lines); the scattering pattern is shown at the right side of the figure. The RBS (Rutherford backscattering spectroscopy) method is illustrated by the projected and the backscattered He⁺ ions beams (gray arrows)

**Fig. 2** Formation of Au-Zn alloy on a QC surface. Cyclic voltammetry of electrodeposition/stripping process of Zn into/from the Au layer on the QC sensor. The inset shows images of the pristine sensor (before cycling) and Au-Zn nanolayer on the surface of the QC sensor (after cycling)

independent RBS measurement for the bare QC), the total mass density of the porous alloy film, $m_p$, was found to be 170.00 ± 1.97 μg/cm². Dividing the calculated mass density by the total porous layer thickness, $h_p = 180.00 ± 3.08$ nm (as measured by AFM), and by theoretical density of the non-porous alloy, $\rho_{np} = 15.01$ g/cm³, the following gravimetric equation was used for calculation of the layer porosity, $\phi$ (defined as the ratio of the volume of pores to the total volume of the porous alloy):

$$\phi = [1 - (m_p/h_p\rho_{np})] \quad (3)$$

The value of $\phi$ was found to be 37.04 ± 1.81% (for details on absolute measurements errors and errors propagation see Supplementary Information Section 6).

**Gravimetric QCM-D measurements in liquids**. The hydrodynamic interaction of a rigid flat surface with liquid causes a negative shift in the $\Delta f/n$ and the coupled positive shift of the resonance width $\Delta W/n$ (both normalized by specific density of liquid, $\rho_{liq}$, and square of the fundamental frequency, $f_0$) as

functions of $\delta$[20,30,31] (the smaller is $n$ the larger is $\delta$ (see Eq. (2)) and the heavier is the effective load of liquid on the QC surface). This linear-type dependence is often called the Kanazawa plot[29]. The QCM-D behavior of non-porous and porous nanolayers in contact with liquids (see Supplementary Fig. 2c, d, respectively) should differ therefore only by the amount of liquid trapped in narrow pores with their widths shorter than the penetration depth. Obviously, the trapped liquid contributes to the $\Delta f/n$ (Fig. 4b) change rather than to any changes in $\Delta W/n$ (Fig. 4a), since dissipation in these systems is only due to the viscous interactions of flat surface with the surrounding external liquid (dissipation is absent in liquid trapped in the narrow pores). In order to separate between the hydrodynamic viscous interaction of the surface of the external nanoporous layer with contacting liquid and the inertial effect of the trapped liquid, the experimental values of $\Delta f/n$ measured in liquid are first referenced to the $\Delta f/n$ values of the layer measured in air. In the second step, the total experimental value of $\Delta f/n$ measured in liquid is corrected by the frequency change due to the viscous interaction of the external surface with the contacting liquid. This two-step correction procedure is further illustrated graphically in Fig. 4.

For bare QCs (consisting of the nonporous Au layer), the experimental QCM-D responses are designated by open black circles in Fig. 4. The QCs containing porous Au-Zn layers were measured in three different liquids: hexane, ethanol and water (open circles with the related error bars, Fig. 4).

The experimental points corresponding to $\Delta W/n$ for nanoporous Au-Zn layers in the three different studied liquids (Fig. 4) ideally fits the straight line for the flat surface (confirming that immersion of the nanoporous film into liquids does not change the normalized shift $\Delta W/n$ with respect to that of the non-porous solid), whereas the corresponding points for $\Delta f/n$ are downward shifted from the linear Kanazawa plot demonstrating a constant shift in the frequency, which does not depend on $n$ for the lower overtone orders ($n$ from 3 to 7). This constant shift in frequency is the QCM-D signature of the inertial load of liquid trapped inside Au-Zn nanoporous layer. The frequency shift (indicated by the vertical bar in Fig. 4) was first transformed into the mass density of the trapped liquid ($m_{liq}$) using Eq. (1), and then porosity, $\phi$, was calculated according to Eq. (4):

$$\phi = m_{liq}/\rho_{liq}h_p \quad (4)$$

Equation (4) contains two experimental quantities, the mean mass density of the liquid (found from 4 independent samples) is

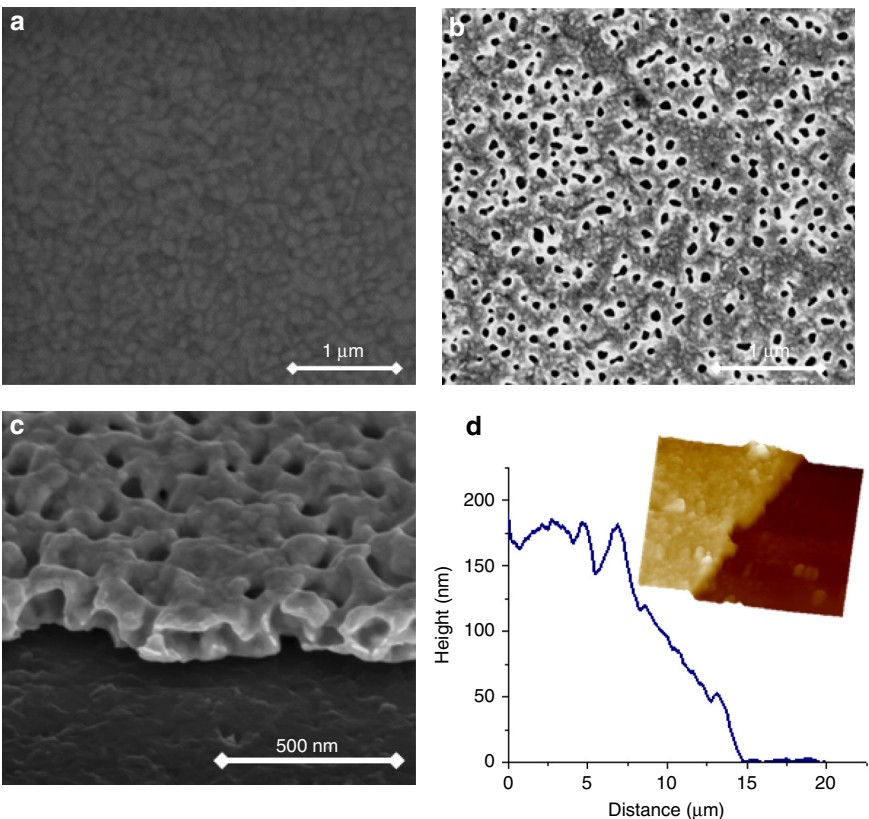

**Fig. 3** Characterization of the Au-Zn nanolayer morphology by SEM. Quartz crystal covered with non-porous Au (pristine neat crystal (**a**)) and nanoporous Au-Zn alloy (**b**), the latter also shown by its tilted image (**c**). The average profile of the nanoporous film measured by AFM is presented in **d**. Inset shows 3D morphological images of the scanned step

$m_{liq} = 6.56 \pm 0.33\ \mu g/cm^2$, and $h_p = 180.00 \pm 3.08$ nm, and the constant value of density of the liquid (water in this case), $\rho_{liq} = 0.997\ g/cm^3$. The value of $\phi$ was found to be $36.60 \pm 2.45\%$, in good agreement with that measured by QCM-D in air, $37.04 \pm 1.81\%$ (for details on the absolute errors and errors propagation, see Supplementary Information Section 6).

For the higher harmonics ($n$ from 7 to 13), the plots in Fig. 4 reveal a characteristic downward and upward deviations of $\Delta f/n$ and $\Delta W/n$, respectively, from the related Kanazawa lines. Since the decaying oscillation amplitude at high $n$ becomes smaller, the acoustic wave effectively probes viscoelasticity of the nanoporous layer when in contact with liquid (it is shown by a more sloping line in Supplementary Fig. 2d).

This conclusion is fully consistent with the fact that the nanoporous Au-Zn layer of a lower porosity ($\phi = 9\%$) attained after the third alloying/dealloying cycle has demonstrated a completely rigid behavior with the trapped liquid inside the nanolayer (see the related SEM image and the normalized plots of $\Delta f/n$ and $\Delta W/n$ vs. $\delta$ in Supplementary Fig. 3a, b, respectively). Hence, the enhanced porosity of the nanoporous alloy formed after seven alloying/dealloying cycles causes an effective softening of the nanoporous layer revealed by the QCM-D measurements in liquids.

**Viscoelastic characterization of nanoporous Au-Zn layers**. The viscoelastic modeling implies a simultaneous fit of the properly referenced experimental changes of $\Delta f/n$ and $\Delta D$ as functions of $n$ to the Voigt-type formula (Supplementary Eq. 3). Several research groups have greatly contributed to the advanced use of concepts of QC admittance for the experimental assessment of raw data and their further modeling[32–35]. The mass density of

the viscoelastic layer expressed by the product of the specific gravimetric density and the thickness of the layer is one of the fitted parameters, in addition to its complex shear modulus.[26] Thus the properly referenced $\Delta f/n$ and $\Delta D$ changes were fitted using the Voigt-type Equation (see Supplementary Note 3 followed by Supplementary Eq. 3).

The fitted parameters are listed in Supplementary Table 1. The electrochemically produced porous Au-Zn layer appears to be viscoelastic with its storage modulus found from fitting the experimental QCM-D responses of four independent samples G' $= 20.00 \pm 1.83$ MPa and loss modulus G'' $= 24.90 \pm 2.45$ MPa (the latter was calculated from the fitted value of the shear viscosity, $\eta_f$, using the formulae: G'' $= (2\pi n f_0)\eta_f$[26]. The fitted value of the density of the nanoporous Au-Zn layer was found to be $9.80 \pm 0.33\ g/cm^3$ (Supplementary Table 1). However, QCM-D measurements of the samples in liquid relate to the average density of the porous host and density of the trapped liquid[16]. Correcting for the contribution of the trapped liquid ($0.366 \pm 0.018\ g/cm^3$, the effective density of the porous layer was found to be $9.43 \pm 0.33\ g/cm^3$ corresponding to $\phi = 37.15 \pm 2.18\%$ vs. $36.60 \pm 2.45\%$ as determined by weighing the trapped liquid in the hydrodynamic approach, and $37.04 \pm 1.81\%$ for the simpler weighing of the nanoporous film in air. This calculation is entirely self-consistent since the parameters of the nanoporous structure of the alloy layers for the measurements in different liquids appeared to be virtually the same (see the quality of the viscoelastic fit in Supplementary Fig. 4 and Supplementary Table 1).

Note that the extensively nanoporous thin Au-Zn alloy films clearly show a viscoelastic behavior when they are in contact with liquids (see QCM-D response for higher harmonics in Fig. 4a, b), whereas same films may be considered as completely rigid

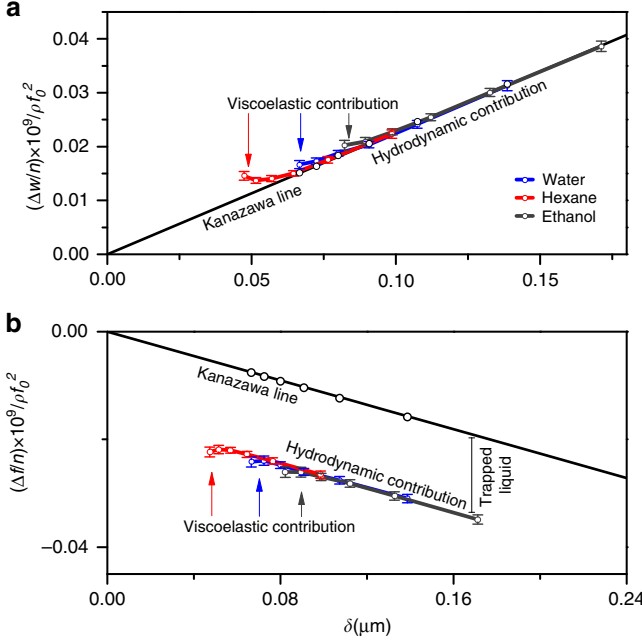

**Fig. 4** QCM-D hydrodynamic characterization of the porous alloy immersed in different liquids. Normalized $\Delta W/n$ and $\Delta f/n$ changes are presented in **a** and **b**, respectively. The reference state is the QC covered by a porous Au layer measured in air. The straight lines reflect the hydrodynamic interactions between the external surface of the solid and the contacting liquids. The contribution of the trapped liquid into the frequency change in liquid is shown by the vertical bar. The viscoelastic contribution to the QCM-D response (i.e., the characteristic deviation of the responses from the straight lines) is seen only in the range of higher overtone orders (for further details, see the text and Supplementary Information Section 6). Error bars relate to standard errors calculated from four independent measurements

during the QCM-D measurements in air (see the $n$-independent difference in the resonance frequency measured with extensively porous Au-Zn films with respect to that of the bare crystal (Supplementary Fig. 6)). To the best of our knowledge, this is the first experimental validation of the principle which we call: the viscoelastic contrast predicted in the theory of QCM-D[26] The viscoelastic correction to the frequency response in air rapidly decreases with the decrease in the films' thickness because the external surface of the films is stress-free, and inertia becomes the sole contribution to the frequency change measured in air. In contrast, the external surface of films attached to QC surfaces is subjected to stress from the surrounding liquid. This makes the viscoelastic contribution to the complex frequency changes much larger than that in air. For this reason, it is perfectly clear that application of the principle of viscoelastic contrast for QCM-D characterization of extensively nanoporous thin films presents a viable and powerful tool of tracking porosity of materials with complex nano-architecture together with their mechanical properties.

**Porosity quantification of the alloy by RBS and GISAXS.** RBS spectra and atomic concentration profiles for bare and porous sensors are presented in Fig. 5. It is seen that after fabrication of the porous layer an additional peak around 1500 KeV was detected, assigned to the formation of Au-Zn alloy, while the peak width of both Au and Cr become wider due to the increase of the layer thickness (in agreement with the AFM data). Nevertheless, the atomic content of Au as well as Cr in the thin films remained

virtually the same (see Supplementary Table 2). The depth profile of the porous sensor (Fig. 5d) shows the presence of Zn throughout all the Au and the Cr layers. Moreover, a deeper penetration of the metallic layers (Au-Zn and Cr) toward the $SiO_2$ substrates was detected. Such observation can originate from mixing between the Au and Zn phases and from the porous nature of the coating, which allows direct path of the He beam to the hidden sublayers, oppositely to the bare sensor.

Considering 1 g of alloy containing 83.2% Au and 16.8% Zn (mass concentration), the bulk density of the non-porous alloy was calculated to be 15.01 $g/cm^3$. This value was found to be in a good agreement with the experimental value for similar Au-Zn composition reported in ref. [36]. The actual density of the nanoporous layer was estimated from the ratio of the measured total mass and the electrode volume: $9.54 \pm 0.32$ $g/cm^3$. Using then the following Eq. (5)

$$\phi = (1 - \rho_p/\rho_{np}) \tag{5}$$

where $\rho_p$ and $\rho_{np}$ are the densities of the porous and the non-porous alloy, respectively, $\phi$ was found to be $36.42 \pm 2.15\%$, in close proximity to the value obtained by QCM-D.

As an additional complementary analysis, $\phi$ was quantified using GISAXS technique based on the ratio of the critical angles for the porous layer and for the bulk Au-Zn alloy ($\alpha_p$ and $\alpha_{np}$ respectively as denoted in Table 1).[37] Details of the technique and data treatment are described in Supplementary Note 4, Supplementary Table 3, and Supplementary Fig. 5. The resulting porosity value, $\phi = 36.8 \pm 1.8\%$ found from the GISAX measurements is in a good agreement with that derived from RBS and from the 3 independent modes of QCM-D we used herein (one for the measurements of nanolayers in air and two in contact with liquids), see Table 1. For more detailed description of the measurement errors and errors propagation during calculation of porosity, see Supplementary Note 5.

## Discussion
A solid scientific background has been developed for tracking and for a quantitative treatment of the potential-dependent dimensional, viscoelastic, and mass changes occurring during charging of battery and supercapacitor electrodes in energy storage devices[38–43]. However, what actually was elaborated in these papers related to tracking of intercalation or adsorption-induced changes in *moderately porous* electrodes rather than determination of *absolute porosity of extensively nanoporous materials*.

The determination of bi-continuous nanoporosity of dealloyed Au-Zn thin films described in the present paper is only one particular example of an intelligent use of the different QCM-D modes (operated in air and in contact with liquids) to obtain self-consistent values of the materials' nanoporosity. There is a vast class of artificial hierarchal extensively nanoporous metallic and non-metallic materials[44] important in the field of catalysis, sensors, fuel cells, batteries, and supercapacitors. Below is a brief list of critically relevant nanotechnological problems of materials properties, which can be much better understood after careful QCM-D studies.

i. The exposure of nanoporous Au films sequentially to $O_3$ and CO gases results in their reversible surface-stress-induced actuation due to the repeated contraction/expansion of the few-nm-size Au filaments[45]. The application of QCM-D for the characterization of nanoporous Au films in these gaseous environments could be challenging for the precise tracking of the in-plane component of the surface-induced stress possibly coupled with the plastic flow.

ii. QCM-D techniques are ideally suitable for assessing critical microstructure evolution of three-dimensional (3D)

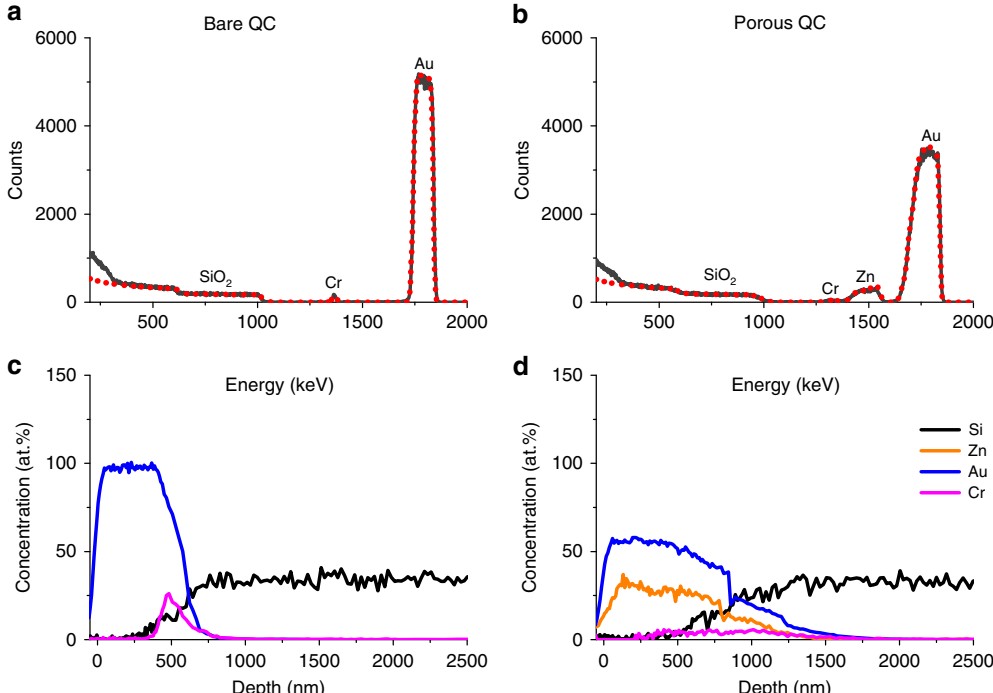

**Fig. 5** Determination of nanoporosity from the RBS spectra. The experimental and simulated spectra are shown by black and dashed red lines, respectively, for the nonporous sensor (**a**) and after the formation of a thin nanoporous layer (**b**). The elemental concentration as a function of probing depth for bare and porous sensors are shown in **c** and **d**, respectively

**Table 1 Combination of three different techniques for porosity determination in thin porous layers**

| Technique | QCM-D/air | QCM-D/hydrodynamic | QCM/viscoelastic | RBS | GISAXS |
|---|---|---|---|---|---|
| Equation | $\phi = [1 - (m_p/h_p\rho_{np})]$ | $\phi = m_{liq}/\rho_{liq}h$ | $\phi = (1 - \rho_p/\rho_{np})$ | $\phi = (1 - \rho_p/\rho_{np})$ | $\phi = 1 - (\alpha_p/\alpha_{np})^2$ |
| Porosity/% | 37.04 ± 1.81 | 36.60 ± 2.45 | 37.15 ± 2.18 | 36.42 ± 2.15 | 36.83 ± 1.80 |

The natural parameters of the porous/viscoelastic structure of the alloys captured by each technique are indicated together with equations for the nanoporosity calculation. The determined values of porosity are listed in the bottom line

nanoporous films produced by dealloying processes. This may result in the entire collapse of the neighboring Au ligaments[46]. Whereas previously QCM-D has been successfully applied for studying structural collapse in soft materials like polymer nanobrushes in liquid environments[47], it has never been used for studying coarsening processes in extensively nano-porous metallic films. The coarsening is believed to occur due to a delicate balance between the surface energy decrease at the expense of the appearing plastic deformation. QCM-D is very well tuned to capture traces of the plastic deformation in addition to tracking the purely elastic effects.

iii. The collapse of extensively nanoporous amorphous Au-Si alloy[48] occurs in a much stiffer solid matrix than that for much softer polymers and biomaterials. The reliable mechanical characterization of the relatively stiff highly porous materials, which represent a vast class of artificial nanostructure-controlled architectures, requires a proper tuning of different QCM-D modes to correctly sense the mechanical state of the moderately stiff materials of complex morphology.

iv. The mechanical deformation of extensively nanoporous films under high strain rate results in their plastic flow caused by abundance of nanovoids in the film[49]. A kind of plastic flow is also observed during the first lithiation of Si anode (Si is one of the most important anode for Li ion

batteries having the highest reversible capacity around 4000 mAh/g) caused by a huge volume change of the intercalation particles (by 350%)[50] and due to the stress experiencing from the side of the electrochemically formed solid–electrolyte interphase[51]. Both the very large volume change and the appearing surface stress are accommodated by a plastic flow. A detailed gravimetric and mechanical QCM-D study of this practically important system presents certainly a challenging research target.

v. Finally, a recent report that dealloying of Ag-Al and Au-Ag alloys can be performed in nonoxidizing acids at ambient temperature and pressure[52] resulting in fabrication of highly nanoporous Ag and Au films will pave the way for the advantageous use of in situ QCM-D in its gravimetric, hydrodynamic, and viscoelastic modes in liquid environment to capture in real time the major critical mechanical events in the films, which determine their stable nanoporous structure assessed in air and in contacting liquids.

In order to tune the advanced QCM-D methods to collecting important complementary information on the nanoporosity of 3D extensively porous materials, the first necessary step was to prove that consistent values of the absolute nanoporosity can be retrieved from all the available QCM-D modes both when measured in air and in contact with liquids. In the second step, the value of porosity obtained from QCM-D studies should be further

validated by the use of two complementary RBS and GISAXS techniques. These rigorous two-step validation process of determination of nanoporosity of extensively porous materials has been well documented in the current paper.

In conclusion, using gravimetric and beyond-the-gravimetric modes of QCM-D for measurements in air and in liquids, we have proposed fast, facile, and convenient method of quantification of nanoporosity and simultaneously viscoelasticity of electrochemically fabricated nanoporous Au-Zn alloys using the principle of viscoelastic contrast. Au-Zn alloy serves as a very good model example of the extensively nanoporous materials. The value of porosity determined by the use of QCM-D in three different modes (one in air and two in liquids) is fully self-consistent and were further validated by the elemental and reflectance data obtained from RBS and GISAXS methods, respectively. Good agreement between the values of porosity obtained by three independent methods using the principle of viscoelastic contrast for QCM-D measurements in air and in liquids provides the necessary proof for the reliability of QCM-D as a powerful method for characterization of extensively nanoporous architectures in which nanoporosity is intrinsically coupled with the mechanical properties of these advanced materials.

## Methods

**Fabrication of a porous Au electrode**. $O_2$ plasma-treated 5 MHz QC sensor (Biolin Scientific, Sweden) was inserted in a home-designed holder and assembled into 3-electrode configuration cell using Zn foil as a counter and reference electrodes. The constructed cell was immersed into 1.5 M $ZnCl_2$ solution in ethylene glycol and then placed on a hot plate and kept at a constant temperature of 120 °C for 10 min prior to and during the electrochemical cycling. The QC sensors are purchased with a nanometric gold layer on top of them. A nanoporous layer of Au-Zn alloy on top of the QC sensor is obtained by a repeated electrochemical alloying/dealloying in 1.5 M $ZnCl_2$ solution that results in the formation of a nanoporous Au-Zn alloy.

Electrochemical procedures and measurements were accomplished with an Autolab potentiostat/galvanostat instrument (Eco Chemie, Netherlands). Porous structures were obtained by application of cyclic voltammetry at a scan rate of 20 mV/s in a potential window between −0.4 V and 1.4 V vs. $Zn/Zn^{2+}$. After every cycling set (as indicated in the paper), the QC was rinsed in ethanol and DD water and dried under pure Ar stream.

**QCM-D measurements**. QCM-D measurements were performed in air and in contact with liquids (water, hexane, and ethanol) before the formation of the porous structures and after the third and the seventh cycling sets, using Q-Sense E1module (QCM-D from Biolin Scientific). The frequency and the dissipation values were simultaneously recorded for all possible overtones from the 3rd to the 13th.

**Morphological visualization and structural information**. These were obtained by high-resolution SEM imaging (Magellan XHR 400 L FE-SEM, FEI) and AFM (Fast Scan Bio atomic force microscope Bruker AXS) at tapping mode with a single FASTSCAN-A silicon probe. Detection of height changes after formation of porous structures was measured by AFM, relative to a non-exposed fraction of the QC periphery (which was covered by Kapton tape during the cycling, in order to avoid involvement of this part of the QC in the electrochemical processes).

**RBS analysis**. Samples for RBS were mounted on the sample holder using double-side, self-adhesive carbon tape. The RBS measurements were performed using a 2.024 MeV $He^+$ ±1 keV beam from the 1.7-MV Pelletron accelerator, NEC. The beam current was ~8 nA, with a nominal diameter of 1.5 mm. The one electron suppressor was used between the beam entrance and the sample holder, biased at −100 V vs ground, and second one that connect in front of sample was biased at −1000 V. Normal incident beam was used in all measurements. All spectra were collected using fixed silicon drift detector (ULTRA$^{TM}$ Silicon-Charged Particle Detector, ORTEC) with 15 keV full width at half maximum. The RBS detector scattering angle, $\theta$, was 169° (Cornell geometry); solid angle, $\Omega$, is 2.7 msr. The NDF code was used to analyze the data. Depth profiles were extracted automatically from RBS spectra using the Surrey IBA DataFurnace software.

**GISAXS measurements**. Experiments were carried out with a small-angle diffractometer (Molecular Metrology SAXS system with Cu Kα radiation from a sealed microfocus tube (MicroMax-002+S) equipped with two Gobel mirrors and three-pinhole slits. The generator was powered at 45 kV and 0.8 mA. The scattering

patterns were recorded by a $20 \times 20$ cm two-dimensional position-sensitive wire detector that is positioned 150 cm behind the sample ($0.075 < h < 2.7$ nm$^{-1}$; $0.1° < 2\theta < 3.7°$). The resolution of the SAXS system is

$$\pi/h_{max} = \pi/2.7 \sim 1.16 \text{nm}$$

where $h = \frac{4\pi \sin\theta}{\lambda}$ is the scattering vector, $\lambda = 1.54$ Å is the wave length of X-ray, and $\theta$ is half of the scattering angle.

## Data availability
The authors declare that all the data supporting the findings of this study are available within the paper and its supplementary information files

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

## Acknowledgements

This work has been partially supported by the Israeli Committee of High Education in the framework of the INREP project and by the Israeli Ministry of Science and Technology and Space Grant 66032. N.S. thanks the Israel Ministry of Science Technology and Space for their financial support.

## Author contributions

Study concept, experimental work and paper writing were equally done by M.D.L., N.S., and D.A. with help from S.S. and F.M. RBS measurements and data processing were carried out by O.G. GISAXS measurements and analysis were conducted by R.L.K.

## Competing interests

The authors declare no competing interests.
