## [Peer Review File · Nature Communications]

Reviewers' comments:

Reviewer #1 (Remarks to the Author):

Responses to my remarks were fully done and correct

Reviewer #2 (Remarks to the Author):

The paper 'Quantification of Nanoporosity in Thin Films in Contact with Gases and Liquids' presents a new approach to measure the porosity in nanoporous Au-Zn thin films based on the quartz crystal microbalance with dissipation monitoring (QCM-D) measurements in air or liquid environments. The results were validated by Brunauer-Emmett-Teller and grazing-incidence small-angle scattering measurements. The viscoelasticity of the thin films was also detected. The paper is well organized, and my previous comments were mostly addressed in this re-submission.

I have thus only a few minor concerns that may be considered by the authors to improve the presentation.

1. The new presentation narrows the study for 'extensively nanoporous films', in which 'nanoporosity is intrinsically coupled with mechanical properties of these advanced materials'. The authors may consider to provide a few examples in the abstract and introductory paragraphs, even the title, for an explicit definition of the term and highlight the importance of the current work.
2. One of the vertical separators for the table in Fig. 6, between 'liquid' and 'vacuum', should be adjusted.

Reviewer #3 (Remarks to the Author):

While we appreciate the changes authors have made including pointing out issues in prior QCM-D works, we do not feel that our previous concerns were fully addressed. Below we detail our concerns with the novelty along with technical questions.

1. For Ref. 14, the authors pointed out two issues: 1) The leaching process might not be complete and 2) The porosity may change once the structure is in contact with liquid. However, it seems that the air method proposed in the current work is still very similar to what was used in Ref. 14 and would not be able to solve these issues either.
2. In the case of Refs. 15, 16, and 17, the authors argued that the air method cannot be used and volume/mass fraction was considered instead of porosity. In the case of Ref. 19, the authors pointed out the failure of the hydrodynamic model. It appears that the framework in these works is very similar to the liquid method proposed in the current work: one combines the dissipation data with the thickness info to obtain porosity. Maybe the author could detail what has been done differently.
3. Given that the air method and the liquid method have been used separately in literature, it appears to us that the main contribution of this work is that the authors showed the air method and liquid method together and showed consistency between the two methods with validation against two reference methods. This is certainly valuable, but unless the authors show the combination of the liquid and gas methods provide important new insight besides the film porosity, the novelty seems incremental.
4. We mentioned in the first round of review *Microporous and Mesoporous Materials* 176 (2013) 71–77 where Thorn et. al. used the contrast between trapped H₂O and D₂O with QCM-D to determine porosity. The authors should probably also discuss this in their literature review section.
5. We asked whether the QCM-D method requires the film to be grown on the quartz crystal in the first round of review. The author did not seem to have addressed the question directly.
6. On the thickness measurement, the authors now stated that it has to be measured separately. But if the thickness cannot be measured in-situ, the porosity value cannot be obtained in-situ. Can the method still be called in-situ?

7. Can the author explicitly define “extensively nanoporous”?

Our response to the second-round comments of reviewers #1-3

Reviewer #1 (Remarks to the Author):

Responses to my remarks were fully done and correct

Our response: we are grateful to this reviewer for his/her previous very helpful comments and generally positive evaluation of our paper.

Reviewer #2 (Remarks to the Author):

The paper ‘Quantification of Nanoporosity in Thin Films in Contact with Gases and Liquids’ presents a new approach to measure the porosity in nanoporous Au-Zn thin films based on the quartz crystal microbalance with dissipation monitoring (QCM-D) measurements in air or liquid environments. The results were validated by Brunauer-Emmett-Teller and grazing-incidence small-angle scattering measurements. The viscoelasticity of the thin films was also detected. The paper is well organized, and my previous comments were mostly addressed in this re-submission.

I have thus only a few minor concerns that may be considered by the authors to improve the presentation.

1. The new presentation narrows the study for ‘extensively nanoporous films’, in which ‘nanoporosity is intrinsically coupled with mechanical properties of these advanced materials’. The authors may consider to provide a few examples in the abstract and introductory paragraphs, even the title, for an explicit definition of the term and highlight the importance of the current work.

Our response: We wish to cordially thank this reviewer for his very extremely good and useful advice. The term “extensively nanoporous films” was introduced into the revised title, abstract, main text and conclusion. We also sharpened the focus on the principle of viscoelastic contrast used in the QCM-D measurements of these films in air and liquids (both terms were defined in details). To the best of our knowledge this is the first demonstration that this important principle allows a viable and powerful tracking of nanoporosity of materials with complex nanoarchitectures linked to their peculiar mechanical properties.

2. One of the vertical separators for the table in Fig. 6, between ‘liquid’ and ‘vacuum’, should be adjusted.

Our response: thanks, corrected.

Reviewer #3 (Remarks to the Author):

While we appreciate the changes authors have made including pointing out issues in prior QCM-D works, we do not feel that our previous concerns were fully addressed. Below we detail our concerns with the novelty along with technical questions.

Our response: We thank this reviewer for his/her extremely strong criticism which finally helped us to formulate very clearly the main intellectual novelty of our paper. The manuscript presents a first experimental verification of extreme usefulness of the principle of viscoelastic contrast during multiharmonic QCM-D characterization of *extensively nanoporous thin films* of complex nanoarchitectures linked to their *unique mechanical properties*. Such deep and consistent analysis of the above link has never been reported neither in the 4 previously published papers (recommended by the reviewer for consideration and comparison) nor in the entire literature, to the best of our knowledge. The definitions of the extensively nanoporous films and the principle of viscoelastic contrast are given in full details not only in the main text but also appear in the revised abstract and included in the new title of the paper (as recommended to us by reviewer # 2 to whom we are very thankful for a good advice).

1. For Ref. 14, the authors pointed out two issues: 1) The leaching process might not be complete and 2) The porosity may change once the structure is in contact with liquid. However, it seems that the air method proposed in the current work is still very similar to what was used in Ref. 14 and would not be able to solve these issues either.

Our response: Based on ref. 14 (in which leaching was not under control) the porosity of nanoporous Au-Zn alloy could never be correctly determined at all until our work reported herein, since we characterized the chemical composition of the de-alloyed product by both RBS and by Faradaic efficiency of the alloying/dealloying process as described in the paper. Tracking the actual composition of the nanoporous films used by us, generalizes the porous structure characterization from a simple single-component metallic film (as presented in ref. 14) to the porous alloys (as demonstrated in our paper).

Although this feature of our work is important it nevertheless does not present the main intellectual novelty of our paper. The novelty is concentrated in two keywords appeared in the revised title and abstract: the use of the principle of *viscoelastic contrast* for complex characterization of porosity and coupled mechanical properties of the *extensively nanoporous films*. We do not accept the reviewer's argument about the use of similar methods in our present work and previous works (see also below).

Regarding the vision of the nanoporosity problem expressed by the reviewer ("the air method proposed in the current work is still very similar to what was used in Ref. 14") we wish to remind that all QCM instruments measure solely the resonance frequency, and all QCM-D instruments measure coupled resonance frequency and dissipation changes only. However, depending on the sequence of the operations during measurements in air and in liquids, and, most importantly, acoustic load impedance modeling, the determined porosity can be either correct or incorrect: once the model is physically unreasonable, the result are, naturally, incorrect. In the present case of the nanoporous alloy the method of ref. 14 will inevitably result in grossly incorrect porosity values whereas our approach has been entirely validated. However, according to the reviewer, the fact that the paper mentioned as ref. 19 in our paper has been published, allegedly questions the novelty claim of our paper. How this attitude of the reviewer can be logically explained? The reviewer in no word criticizes our experiments, way of modeling, and link between the measurements in air and in liquids.

2. In the case of Refs. 15, 16, and 17, the authors argued that the air method cannot be used and volume/mass fraction was considered instead of porosity. In the case of Ref. 19, the authors pointed out the failure of the hydrodynamic model. It appears that the framework in these works is very similar to the liquid method proposed in the current work: one combines the dissipation data with the thickness info to obtain porosity. Maybe the author could detail what has been done differently.

Our response: Our answer is similar to that given for the previous reviewer's comment: All QCM-D instruments measure coupled resonance frequency and dissipation changes from which the determined porosity appears to be either correct or incorrect depending on the adopted model of the acoustic load impedance. The major fault of ref. 19 is that the link between the porosity and the mechanical properties of the material was grossly incorrect because of the physically unreliable model. The combination of a viscoelastic film with its strong roughness accounted for by the hydrodynamic model is absolutely meaningless (as already explained by us in the previous response to this reviewer) resulting in meaningless porous structure parameters. The correct solution of the problem of porosity of *extensively nanoporous films* affecting their mechanical properties is presented in our paper. Our analysis is based on the principle of *viscoelastic contrast* during QCM-D measurements in air and in liquids. Frankly speaking, the negative attitude of this reviewer concerning our novelty claims activated us in a precise formulation of the intellectual novelty of the paper.

It is highly important to emphasize that our analysis was fully validated by external analytical efforts based on RBS and XRD.

3. Given that the air method and the liquid method have been used separately in literature, it appears to us that the main contribution of this work is that the authors showed the air method and liquid method together and showed consistency between the two methods with validation against two reference methods. This is certainly valuable, but unless the authors show the combination of the liquid and gas methods provide important new insight besides the film porosity, the novelty seems incremental.

Our response: We agree with the first claim of the reviewer (about consistency of methods and results) and disagree with his/her second statement. It can be one simpler case when absolutely rigid porous film is measured in air and in liquid: subtracting the frequency change measured in air from the frequency change measured in liquid separates the hydrodynamic response in liquid which together with the related dissipation changes allows to characterize porous structure of the films via hydrodynamic modeling. It is quite a different case when extensively nanoporous layers behave as rigid in measurements in air but nevertheless, its porosity and viscoelastic properties still can be assessed from the measurements performed in liquids. We call this the principle of *viscoelastic contrast* (fully defined in the revised paper). This is our authentic response to the reviewer's guess that (may be) our "combination of the liquid and gas methods provide important new insight besides the film porosity". Yes, it is fully provided (see details in the revised paper). We provide a first experimental verification of the principle of viscoelastic contrast (predicted by QCM-D theory but yet not verified experimentally)

which opens the way for fast real time characterization of extensively nanoporous thin films of complex nanoarchitectures.

4. We mentioned in the first round of review *Microporous and Mesoporous Materials* 176 (2013) 71–77 where Thorn et. al. used the contrast between trapped H₂O and D₂O with QCM-D to determine porosity. The authors should probably also discuss this in their literature review section.

Our response: We have included this paper into the introduction with our short comment on it.

5. We asked whether the QCM-D method requires the film to be grown on the quartz crystal in the first round of review. The author did not seem to have addressed the question directly.

Our response: Five practically important nanoporous materials are listed in the section of the revised manuscript “Gravimetric and beyond-the-gravimetric QCM-D approach to solving important nanotechnology problems of new functional materials”. The characterization of these materials implies different methods of attachments of the related samples to the surfaces of quartz crystals. In our practice, we attached the electrode particles without binder by spray-pyrolysis, electrophoretic deposition, spraying of dispersions containing binders, and many others. All these methods were included in our last review published during the reviewing process of the present manuscript (Netanel Shpigel, Mikhael D. Levi, Doron Aurbach, *Energy Storage Materials*, Available online 26 May 2019 In Press, Corrected Proof).

6. On the thickness measurement, the authors now stated that it has to be measured separately. But if the thickness cannot be measured in-situ, the porosity value cannot be obtained in-situ. Can the method still be called in-situ?

Our response: The most fundamental fact of the theory and practice of gravimetric QCM-D is that the method (by itself) deals with the so-called Sauerbrey thickness rather than physical thickness of the attached films. Consequently, for reliable measurements of porosity of thin films in air and in liquid, the thickness should be obtained by one of the complimentary techniques. Why thickness of a thin film cannot be measured in situ by AFM? In our practice we always measure thickness *in-situ* by AFM in parallel to the QCM-D measurements. The only problem may be an inhomogeneity of the films. For homogeneous films there is not a problem at all.

7. Can the author explicitly define “extensively nanoporous”?

Our response: This is explained in the introductory response and in the main text of the paper.

REVIEWERS' COMMENTS:

Reviewer #1 (Remarks to the Author):

I consider again that my remarks and comments were fully successful.

Now, about the comments of reviewer 3, reading the responses of the authors, I think that the main authors' comments can be considered as correct and well fitted.

This paper can be published after this last turn of corrections.